# The Pleiotropic Effect of Complement C5a-C5aR1 Pathway in Diseases: From Immune Regulation to Targeted Therapy

**DOI:** 10.3390/ijms262311693

**Published:** 2025-12-03

**Authors:** Baohong Xu, Zhi Zhou, Yang Xiao, Qiaolin Liu, Tiaoyi Xiao, Zhao Lv, Hongquan Wang

**Affiliations:** 1Fisheries College, Hunan Agricultural University, Changsha 410128, China; 2Hunan Engineering Technology Research Center of Featured Aquatic Resources Utilization, Hunan Agricultural University, Changsha 410128, China

**Keywords:** complement system, C5a-C5aR1 pathway, targeted therapy, cancer

## Abstract

The complement system plays a pivotal role in the body’s immune defense mechanism. Its key effector molecule C5a and its primary receptor C5aR1 exhibit complex “double-edged sword” effects in various disease processes, including infectious diseases, inflammatory conditions, tumors, and metabolic disorders. Under normal physiological conditions, moderate levels of C5a bind to the C5aR1 receptor, recruiting immune cells to the site of infection to participate in immune defense and enhancing the body’s ability to clear pathogens. However, in various disease states, the C5a-C5aR1 pathway primarily shapes the disease microenvironment through regulating cellular pro-inflammatory and immune functions, angiogenesis processes, and tissue repair processes. It also promotes tumor immune escape through a novel mechanism through modulating the polarization of myeloid-derived suppressor cells (MDSCs) and regulating T cell function. The C5a-mediated “inflammation–fibrosis–metabolic reprogramming” vicious cycle has become a key molecular basis driving disease progression, maintaining pathological states, and promoting abnormal tissue damage repair in chronic inflammatory diseases. Through elucidating the structural biology of C5aR1 and designing allosteric modulators, nanobodies, and bifunctional molecules as new targeted intervention strategies, we aim to accelerate research progress in related medical fields. This article reviewed the molecular mechanisms of the complement system in tumor immune escape, chronic inflammation, fibrosis, and cardiovascular diseases, and explored the translational potential of targeted interventions. These discussions provide a solid theoretical foundation and new research perspectives for the medical field, aiding in the advancement of further discoveries.

## 1. Introduction

Complement system is a complex, dynamic, and ubiquitous system of more than 50 sequentially arranged proteins that forms a critical arm of immune system [1]. Although initially considered to be only an important component of innate immune system, research has shown that complement system has a greater functional relevance, participating in novel functions such as maintaining homeostasis, development, and regulating synaptic pruning [1,2,3]. Additionally, complements are known to bridge the innate and adaptive immune systems in sterile inflammation [4]. With current high incidence of metabolic diseases, degenerative diseases, and malignant tumors closely related to chronic inflammation [5,6,7,8,9,10], the role of complement system in inflammation regulation has once again gained attention [11,12,13,14,15].

The three major activation pathways (classical, lectin, and alternative) of complement system have been extensively described, and the roles of their products in immune regulation and other biological processes have also been widely reviewed in several excellent review articles (Figure 1) (for instance, [1,3,4,12,16]). During the process of complement activation, complement fragments C3a and C5a generated from the proteolytic cleavage of complement C3 and C5, respectively, also known as anaphylatoxins, play a central role in response to complement activation [17,18]. These anaphylatoxins play important roles in the pathogenesis of allergy, autoimmunity, neurodegenerative diseases, cancer and infections through ligation of their receptors, i.e., the C3a receptor (C3aR), C5a receptor 1 (C5aR1), and the C5a receptor-like C5L2 (C5aR2) [17].

As an important product in the process of complement activation, considerable progress has been made in the research on the role of C5a in immune regulation and health [19]. Under normal physiological conditions, moderate levels of C5a bind to the C5aR1 receptor, recruiting immune cells to the site of infection to participate in immune defense and enhancing the body’s ability to clear pathogens. For instance, in the early stages of acute kidney infection, the C5a-C5aR1 pathway promotes leukocyte recruitment, aiding in the defense against invading pathogens [19]. However, in various disease states, this pathway also contributes to disease progression. In ducklings infected with *Riemerella anatipestifer*, C5a-C5aR1 exacerbates tissue damage through upregulating pro-inflammatory factors such as TNF-α and IL-6. Conversely, targeted blocking of this pathway significantly reduces mortality in infected ducklings and effectively alleviates inflammatory responses [20]. In colorectal cancer cells, through the intracellular complement activation mechanism, cells utilize cathepsin D to cleave C5 into C5a, which then stabilizes β-catenin protein via the KCTD5/cullin3 complex, ultimately driving Wnt pathway-dependent carcinogenesis [21]. One notable issue related to C5a/C5aR1 is that its biological activity is greatly reduced when the carboxyl-terminal residue (Arg) is removed by carboxypeptidases, resulting in C5a-desArg [22]. Although C5a-desArg has many of the same functions as C5a, higher concentrations are needed to elicit biological responses [22]. Furthermore, C5a-desArg may differ from C5a in its activity [22]. Therefore, a systematic review of the progress of research on the roles of C5a in diseases will help us to understand its value and risk as an immune regulation target. Therefore, this paper reviewed the recent progress in the functional research of C5a and evaluated the feasibility of the C5a-C5aR1 pathway as a target for disease treatment.

## 2. Molecular Structure and Signal Transduction Characteristics of C5a-C5aR1

In complement system, the C5a-C5aR1 pathway is a crucial inflammatory regulatory pathway [17]. This pathway is formed by the interaction between complement fragment C5a and its specific receptor C5aR1 (CD88). C5a is a glycosylated peptide produced by the cleavage of C5 through C5 convertase (such as C3bBbC3b and C4b2a3b) (Figure 2). Human C5a consists of 74 amino acids and is the most potent inflammatory anaphylatoxin [23,24]. It is a potent agonist for myeloid cells, particularly neutrophils, which express high levels of C5aR1 and respond chemotactically, as well as having additional biological responses. Non-myeloid cells, such as bronchiolar and alveolar epithelial cells and endothelial cells, have also been shown to express C5aR1 and respond to C5a [25]. C5a not only recruits immune cells such as monocytes, granulocytes, and mast cells to the site of infection, but also induces smooth muscle cell contraction, vasodilation, granulocyte, and mast cell degranulation, and promotes cytokine secretion [26]. In the complement cascade reaction, C5a plays a central role as the terminal effector molecule. This molecule binds to C5aR1 to trigger a series of biological effects, playing a broad and critical role in regulating inflammation and immune response. It influences the body’s immune microenvironment and the progression of inflammatory responses through mechanisms such as activating immune cells and regulating cytokine secretion [24].

High-resolution cryo-electron microscopy technology has successfully resolved the three-dimensional structures of the complexes formed between C5aR1 and its natural ligand C5a, as well as between C5aR1 and the biased agonist BM213. The phosphorylation modification and internalization process of the receptor’s C-terminal region are closely associated with the conformational rearrangement of helix 8 (H8) [28]. This discovery provides crucial structural evidence for our deeper understanding of the bidirectional regulation of C5aR1 in acute inflammatory responses and immune tolerance balance, and for the first time reveals the dynamic conformational changes occurring in transmembrane domain 7 (TM7) and H8 during receptor activation [29]. The confirmatory biophysical validation of C5a “bipartite” binding interaction highlights the energy importance of the aspartic acid in the NT peptide of C5aR1 for C5a binding [30].

C5aR1 is a classical G protein-coupled receptor (GPCR) that is expressed in macrophages, neutrophils, and T cells [31,32]. Additionally, C5aR1 can be induced to express in non-immune cells, such as endothelial cells and neurons. C5a and lipopolysaccharide (LPS) can synergistically induce the production of cytokines and chemokines in various cell types. For example, C5a can stimulate the release of IL-1 and TNF from mouse peritoneal macrophages and human monocytes [33]. This release of inflammatory mediators, including leukotrienes and prostaglandins, further exacerbates the inflammatory response, and significantly enhances the body’s ability to clear pathogens (Figure 3).

These findings not only clearly elucidate the allosteric regulatory mechanisms of C5aR1 multi-signal transduction, but more importantly, they lay a solid structural biology foundation for the development of targeted antagonists for diseases such as rheumatoid arthritis. Additionally, C5aR1 and complement receptor C5L2 (C5aR2) exhibit certain interactions within cells. Although this does not directly indicate the formation of a functional heterodimer between the two, it undoubtedly provides highly valuable research clues for this potential possibility [34]. C5aR2 regulates neutrophil activation and function contributing to neutrophil-driven epidermolysis bullosa acquisita [35]. Its activation is reported that broadly modulates the signaling and function of primary human macrophages [36]. Moreover, it promotes protein kinase R expression and contributes to NLR family pyrin domain-containing 3 (NLRP3) inflammasome activation and the proinflammatory protein high-mobility group box 1 (HMGB1) release from macrophages [37]. Although considerable progress has been made in recent years regarding the role of C5aR2 in immune regulation, and this has been summarized in excellent reviews [38,39], its function remains highly controversial.

**Figure 3 ijms-26-11693-f003:**
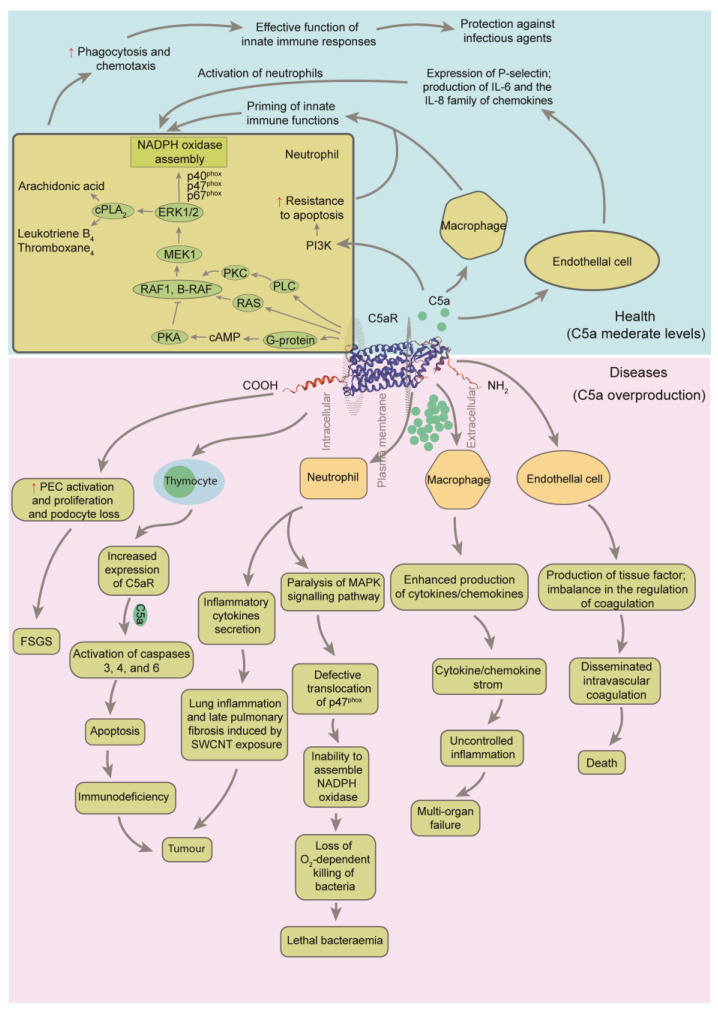
Regulatory framework of C5a on the immune system after combination to C5a receptor 1 (C5aR1) [25,40,41]. Ligation of C5aR1 on neutrophils with C5a results in activation of phospholipase C (PLC), which further activates protein kinase C (PKC). PKC strongly activates rat sarcoma protein (RAS), rapidly accelerated fibrosarcoma proto-oncogene serine/threonine-protein kinase (RAF1) and B-Raf proto-oncogene, serine/threonine kinase (B-RAF). This is followed by activation of mitogen-activated protein kinase kinase 1 (MEK1), resulting in the phosphorylation of p42 and p44 subunits of extracellular signal-regulated kinase 1/2 (ERK1/2). Activated ERK1/2 causes phosphorylation of the cytosolic subunits of NADPH oxidase, p40^phox^, p47^phox^ and p67^phox^. The final outcome of these events is translocation of cytosolic subunits to the cell membrane, where NADPH oxidase is assembled. Phosphorylation of p47^phox^ is blocked when neutrophils are incubated with high concentrations of C5a. cPLA_2_, cytoplasmic phospholipase A_2_; PI3K, phosphatidylinositol 3-kinase; MAPK, mitogen-activated protein kinase; SWCNT, single-walled carbon nanotube. Gray arrows indicate the regulation processes. The red up arrows indicate that it is up-regulated or strengthened.

## 3. Dual-Regulatory Roles of C5a-C5aR1 Pathway in the Tumor Microenvironment

The C5a-C5aR1 pathway exhibits bidirectional effects on tumor behavior. On one hand, it can promote tumor growth by creating an immunosuppressive microenvironment that hinders the immune system’s ability to detect and eliminate tumor cells. It can also stimulate the formation of new blood vessels, providing necessary nutrients for tumor growth and spread [42]. However, under specific conditions, this pathway can also activate the anti-tumor immune response. The activation and function of the C5aR1 pathway exhibit spatiotemporal heterogeneity. Different stages of tumor development and different areas within the tumor tissue can result in varying levels of activation and different modes of action for this pathway. Additionally, the concentration of the C5a ligand plays a crucial role in regulating the effects of the C5a-C5aR1 pathway. Different levels of C5a binding to C5aR1 can trigger distinct downstream pathways, resulting in markedly different biological outcomes. The type of tumor and the stage of disease progression are also important factors that contribute to the diverse regulatory effects of the C5a-C5aR1 pathway.

### 3.1. C5a-C5aR1 Pathway Promotes Tumorigenesis

Abnormal complement activation has been observed in patients with tumors. In lung cancer patients, plasma C5a levels are significantly elevated [43]. Similarly, in breast cancer, high expression of C5aR1 is associated with larger tumor size, lymph node metastasis, and advanced clinical stages. Patients with high C5aR1 expression have lower survival rates compared to those with low expression [44]. C5aR1 levels have also been linked to prognosis in patients with gastric, ovarian, and urothelial carcinoma [45]. Additionally, the complement system is widely activated in colorectal cancer [21].

The presence of C5a weakens the anti-tumor immune response of CD8^+^ T cells and creates a favorable environment for tumor cells to escape immune surveillance. When C5a specifically binds and activates C5aR1, it triggers a series of key biological events. On one hand, this pathway upregulates the expression of cytokines with immunosuppressive functions, such as interleukin 10 (IL-10), and transforming growth factor β (TGF-β), in the tumor microenvironment. On the other hand, it strongly inhibits the normal transduction of proinflammatory pathways, such as IL-12. This leads to a shift in the tumor microenvironment from an immune-balanced state to an immunosuppressive state, ultimately promoting tumor progression. However, when the concentration of C5a is too high, it can lead to excessive activation of immune cells, causing an imbalance in the inflammatory response and an increase in the release of immunosuppressive molecules, which is not beneficial for anti-tumor immunity. For instance, in some inflammation-related tumors, sustained and excessively high levels of C5a can lead to fatigue or exhaustion of immune cells, reducing their ability to kill tumor cells. In cases where tumor cells are able to secrete C5a, it can upregulate the expression of chemokine MCP-1, and promote the metastasis of colon cancer cells to the liver through inflammatory infiltration mediated by MCP-1. In a mouse model of colon cancer liver metastasis, the expression level of C5a is also significantly increased. This leads to an increase in the expression of MCP-1, promoting the accumulation of macrophages and neutrophils in the tumor microenvironment and creating a pro-metastatic inflammatory microenvironment. This enhances the migratory and invasive capabilities of tumor cells, promoting tumor liver metastasis [46]. The upregulation of C5a efficiently recruits macrophages and neutrophils, which then release various cytokines, proteases, and reactive oxygen species, gradually establishing a pro-metastatic inflammatory microenvironment. In such a microenvironment, the migration and invasion capabilities of tumor cells are significantly enhanced. Therefore, key proteins in the C5a-C5aR1 pathway hold promise as important biomarkers for early screening of colorectal cancer. Detecting the expression levels of C5a, C5aR1, KCTD5, and β-catenin in feces or blood can aid in the early detection and treatment of cancers, significantly improving patient survival rates [47].

Blocking C5aR1 inhibits progress in tumor metastasis. Deficiency in C5aR1 has been shown to reprogram macrophages from a pro-tumor state to an anti-tumor state, resulting in increased immune responses and stimulatory pathways. This leads to enhanced anti-tumor activity by cytotoxic T cells, which is dependent on the chemokine (C-X-C motif) ligand 9 (CXCL9). Furthermore, pharmacological inhibition of C5aR1 has been found to improve the efficacy of immune checkpoint blockade therapy. This study reveals the mechanism by which the C5a-C5aR1 pathway regulates tumor-associated macrophage (TAM) anti-tumor immune responses and highlights the potential of targeting C5aR1 for clinical applications [48].

The mechanisms by which C5a-C5aR1 promotes tumor immune escape include: Firstly, C5a recruits myeloid-derived suppressor cells (MDSCs) to the tumor site via C5aR1, where they secrete IL-10, TGF-β, and other cytokines that suppress T cell function. This also induces MDSCs to express ARG1 and iNOS, depleting local arginine levels and inhibiting CD8^+^ T cell proliferation [21,49]. Secondly, C5aR1 activates the expansion of Tregs, inhibiting effector T cell activity through the CTLA-4 and PD-1 pathways [50]. Thirdly, C5a drives TAMs to polarize towards the M2 type, resulting in the secretion of VEGF, MMP-9, and other pro-angiogenic and pro-metastatic factors [51]. Additionally, C5a/C5aR1 not only promotes the formation of a tumor immunosuppressive microenvironment, but also participates in tumor angiogenesis and activates tumorigenic pathways within tumor cells. Currently, various new drugs targeting C5a-C5aR1 are developing, and studies on their use alone or in combination with other immunotherapies for tumor treatment are gradually becoming a hot topic. The C5aR1 has emerged as a new class of immune checkpoint receptor in tumor immunotherapy [52].

### 3.2. Antitumor Potential of the C5a-C5aR1 Pathway

By mediating interactions between cells and between cells and the extracellular matrix, C5a plays a crucial role in both immune and non-immune functions in both plasma and the interstitial tissues outside of blood vessels. In addition to its immune regulatory functions, C5a has been shown to have a significant impact on the plasma levels of IL-17F in sepsis mice induced by LPS- and CLP, with higher levels of C5a correlating with elevated levels of IL-17F [53]. At lower concentrations, C5a primarily binds to C5aR1 on the surface of immune-activated cells, such as dendritic cells (DCs), to promote the transmission of immune activation signals and enhance antitumor immune capacity. The multifaceted roles of C5a-C5aR1 in tumor immunity include: First, direct cytotoxic effects. High concentrations of C5a can induce tumor cell apoptosis by activating the JNK/p38MAPK pathway downstream of C5aR1, triggering the mitochondrial apoptosis pathway. In certain types of tumors, such as melanoma, C5a may also enhance antitumor immunity through recruiting neutrophils. Second, immune cell activation effects. C5aR1 activation can directly trigger the activation of NKT and NK cells, with NKT cells guiding NK cells to the site of infection and forming a unique cytokine profile. Additionally, γδT cells act as antigen-presenting cells, bridging the gap between innate and adaptive immunity [54,55]. Third, synergistic effects with radiotherapy and chemotherapy. Local complement activation, such as after radiotherapy, can recruit immune effector cells via the C5a-C5aR1 pathway, enhancing tumor antigen release and promoting immunogenic cell death [56].

The dual role of the C5a-C5aR1 pathway and its complex interactions with immune cells provide new targets for elucidating the mechanisms of tumor metastasis and developing clinical treatment strategies. In-depth exploration of its regulatory mechanisms is of critical importance for the development of precise and effective anti-tumor metastasis therapies.

## 4. C5a-C5aR1 Pathway Regulates Fibrosis Processes

Renal interstitial fibrosis is a pathological process that can lead to end-stage renal failure in various chronic kidney diseases. In patients with IgA nephropathy, the activation of C5aR1 can trigger renal epithelial–mesenchymal transition (EMT), which ultimately leads to renal interstitial fibrosis. As the severity of renal interstitial fibrosis worsens, the expression of C5aR1 also increases. This increase in C5aR1 expression has been found to be correlated with an increase in Snail factor expression, which can activate the EMT process and contribute to the development of renal interstitial fibrosis [57]. Additionally, Liu et al. [58] discovered that components of the complement system are involved in the damage to renal tubulointerstitial tissue. Specifically, C3a and C5a have been found to induce the expression of β-catenin mRNA and protein in renal tubular epithelial cells, thereby affecting the Wnt/β-catenin pathway and contributing to the phenotypic transformation of these cells. However, the effects of C3a and C5a on β-catenin can be attenuated by C3a and C5a receptor antagonists, respectively. This is important because renal tubular epithelial cells express surface receptors for C3a and C5a, which can upregulate the interaction between tissue C3a and C5a and their respective receptors, leading to the activation of downstream pathways and ultimately inducing EMT. In vitro studies have shown that C3a and C5a can promote renal tubulointerstitial fibrosis and induce EMT in renal tubular epithelial cells, suggesting that these components may play a role in the EMT process by binding to their receptors (C3aR and C5aR1), and causing EMT in renal tissue, and ultimately leading to renal interstitial fibrosis. Clinical studies have also found a significant positive correlation between C5a levels in patients and the severity of renal interstitial fibrosis pathology, further supporting the idea that the C5a-C5aR1 pathway plays a key promotional role in the onset and progression of this condition. Therefore, targeting the C5a-C5aR1 pathway may be a promising approach for the diagnosis and treatment of renal interstitial fibrosis.

## 5. The C5a-C5aR1 Pathway Regulates Chronic Inflammation Mechanisms in Vascular Calcification Processes

Vascular calcification is a common pathological change in cardiovascular diseases, significantly impairing vascular elasticity and function whereas increasing the risk of cardiovascular events. Liu et al. [58] comprehensively elucidated the key regulatory mechanisms of the C5a-C5aR1 pathway in the vascular calcification process. They found that when C5a specifically interacts with the C5aR1 receptor on the surface of vascular smooth muscle cells (VSMCs), it triggers an endoplasmic reticulum stress (ERS) response. This, in turn, activates the PERK-eIF2α-ATF4-CREB3L1 pathway. Under normal physiological conditions, the endoplasmic reticulum (ER) serves as a critical site for protein processing within the cell. This includes signal sequence cleavage, N-linked glycosylation, disulfide bond formation, isomerization or reduction (catalyzed by protein disulfide isomerase (PDIS) and oxidoreductases), proline isomerization, or lipid conjugation. These processes ultimately result in the correct folded formation of proteins [59,60,61,62]. However, conditions that disrupt endoplasmic reticulum homeostasis can induce a cellular state commonly referred to as “ER stress”. The cellular response to ER stress includes activating adaptive mechanisms to overcome stress and restore endoplasmic reticulum homeostasis. This response depends on the type and intensity/duration of the disruptor/condition [63]. Additionally, adverse stimuli can also activate the complement system, leading to excessive production of C5a and its binding to the cell surface C5aR1. The activation of this pathway forms a complex interactive network with ERS, jointly participating in pathological processes [64].

These findings not only deepen our understanding of the pathological mechanisms underlying vascular calcification, but also provide potential intervention targets for developing novel therapeutic strategies targeting this disease. For instance, using RNAi technology to knockdown PERK has demonstrated that the PERK pathway of endoplasmic reticulum stress plays a role in the high-glucose-induced transdifferentiation of VSMCs into osteoblast-like cells. By interfering with PERK, this transdifferentiation process can be inhibited, which is beneficial for intervening in the pathophysiological phenomenon of vascular calcification and can serve as a target for early prevention of vascular calcification. Additionally, the development of antagonists targeting the C5a-C5aR1 pathway has been supported, such as using anti-C5a monoclonal antibodies to bind C5a and block this pathway. This can effectively reduce inflammatory responses and provide new insights for treating inflammatory and autoimmune diseases. These research advancements also bring new hope for conquering vascular calcification-related diseases and will further drive the progress of translating basic research into clinical applications in the future.

## 6. The C5a-C5aR1 Pathway in the Autoimmune Pathophysiology of Rheumatoid Arthritis

Rheumatoid arthritis (RA) is an autoimmune disease characterized by symmetrical polyarthritis. Its pathological features include synovitis, which involves synovial hyperplasia, vascular proliferation, and immune cell infiltration. Although many T cells are involved in the pathogenesis of RA, recent studies have highlighted the role of T helper cell 17 (Th17) and their cytokines in synovitis, cartilage destruction, and bone erosion/destruction in RA. By understanding the mechanisms of Th17 cells and their cytokines, new therapeutic strategies for RA and other autoimmune diseases have been proposed [65]. The C5a-C5aR1 pathway plays a central role in disrupting immune balance through regulating the functions of Th17 and regulatory T cells (Tregs). When C5a binds to C5aR1, it significantly enhances the differentiation and function of Th17 cells. Th17 cells are a subset of T helper cells that produce interleukin-17 (IL-17) and play a crucial role in inflammation and tissue damage. As a new type of T helper cell lineage, Th17 is associated with numerous autoimmune diseases [66,67,68,69]. Additionally, the C5a-C5aR1 pathway inhibits the function of Tregs. A reduction in Treg numbers or impaired function prevents them from effectively exerting immune suppressive effects, leading to the breakdown of immune tolerance towards self-tissues and further exacerbating immune imbalance [66,67,68,69]. During the pathogenesis of RA, the C5a-C5aR1 pathway primarily disrupts immune balance through regulating the functions of Th17 and Treg. Therefore, C5a is the primary product of complement activation responsible for tissue damage in RA, although the deposition of membrane attack complexes and the opsonic effects of C3b fragments are also important. The success of complement inhibition in experimental models has encouraged the development of new therapeutic approaches for human RA [70].

## 7. The C5a-C5aR1 Pathway Participates in the Pathogenesis and Regulation of Metabolic Diseases

Metabolic-associated fatty liver disease (MAFLD) is a chronic metabolic dysfunction-related liver injury characterized primarily through excessive fat accumulation in the liver. It was renamed from nonalcoholic fatty liver disease (NAFLD) in 2020 [71]. Preliminary estimates suggest that there are over 200 million MAFLD patients in China, and this number is predicted to exceed 300 million by 2030 [72]. The C5a-C5aR1 pathway, a key hub connecting metabolic disorders and inflammatory fibrosis, plays an important role in promoting the development of MAFLD through participating in the pathological processes at various stages through multidimensional regulation [73,74].

The activation of the C5a-C5aR1 pathway is triggered through three major types of stimuli. Free fatty acids (FFAs), such as palmitic acid and oleic acid, can directly damage liver cell membranes, exposing danger signals like phosphatidylserine, thereby initiating the classical complement pathway. Additionally, lipid overload induces hepatocyte apoptosis, with apoptotic bodies activating the classical pathway via C1q to generate the inflammatory mediator C5a. Weakened insulin signaling inhibits the expression of complement regulatory proteins, such as CD55 and CD59, lifting the negative regulation of complement activation. Reactive oxygen species (ROS) generated through oxidative stress modify low-density lipoprotein (LDL), forming oxidized LDL (ox-LDL), which activates the classical complement pathway via C1q. Clinical studies have shown that serum C5a levels in MAFLD patients are positively correlated with alanine aminotransferase (ALT), aspartate aminotransferase (AST), and insulin resistance index (HOMA-IR). The expression of C5aR1 in liver tissue is significantly correlated with fibrosis stages (Ishak score) and NASH activity score (NAS). In patients with NASH and fibrosis, the density of C5aR1-positive cells (HSCs and macrophages) is 2–3 times higher than in patients with simple fatty liver, suggesting that the activity of the C5a-C5aR1 pathway is closely related to disease progression.

## 8. Current Status of C5a/C5aR1 Targeted Therapy Research

The complement system, as one of the primary components of the innate immune system, plays a crucial role in protecting the body against pathogens, trauma, and altered host environments [75]. However, dysregulation of the C5a-C5aR1 pathway is associated with a myriad of acute and chronic inflammatory conditions and neurodegenerative diseases [76]. Therefore, the therapeutic inhibition of this pathway is imperative for the treatment of these abnormalities [76]. Peptidic PMX53, PMX205, and JPE1375, and non-peptidic W545011, NDT9513727, DF2593A, and CCX168 were the most commonly reported and clinically advanced small-molecule C5aR1 inhibitors [76]. They vary widely in their pharmacological profiles and have been characterized using different cell models and experimental protocols, which lead to inconsistent conclusions. For instance, DF2593A demonstrated apparent nanomolar potencies on both human and rodent C5aR1s and significant antinociceptive effects in several models of inflammatory pain [77]. However, Li et al. [76] reported that DF2593A failed to demonstrate any significant inhibition of C5a signalling. Moreover, through signalling assays measuring C5aR1-mediated cAMP and ERK1/2 signalling, and β-arrestin 2 recruitment, Li et al. [76] demonstrated that the high insurmountable antagonistic potencies for the peptidic inhibitors as compared to the non-peptide compounds in primary human macrophages, although the peptidic inhibitors have the low oral bioavailability, short plasma half-life and high synthetic costs [78]. Nevertheless, research on C5a/C5aR1 targeted therapy has still made commendable progress. Recently, a study using an LPS-induced mouse model showed that C5aR blockade with the C5aR antagonist W54011 reversed LPS-induced acute kidney injury and attenuated mitochondrial damage in tubular epithelial cells [79]. Being orally bioavailable and with a good biosafety feature, CCX168 (avacopan) has passed Phase III clinical trials and has been approved for the treatment of antineutrophil cytoplasmic antibody-associated vasculitis [80,81].

Based on the dual role mechanism of the C5a-C5aR1 pathway in tumor immunity and its potential for clinical translation, related therapeutic strategies are gradually transitioning from basic research to diversified areas. These strategies include: 1. Antagonist development and clinical trials. a. PMX-53 and avdoralimab can block the C5a-C5aR1 pathway, reducing the infiltration of MDSCs and Tregs. In a melanoma mouse model, the activation of autocrine C5aR1 on tumor-infiltrating CD8^+^ T cells altered their antitumor activity and promoted cancer progression. This suggests that combining C5aR1 inhibitors with immunotherapy may have potential benefits [82,83]. b. Bias antagonists encompass a broad range of knowledge in the GPCR field, where β-arr integrates signals generated through GPCR with intrinsic cellular pathways in human diseases. This initiates intracellular signaling waves in a G protein-independent manner, enabling the discovery of new therapies that selectively target β-arr-mediated circuits [84]. 2. Combination with immune checkpoint inhibitors. a. Reversing T cell exhaustion. C5aR1 inhibitors reduce the release of immunosuppressive factors, such as IL-10 and TGF-β, in the tumor microenvironment. This restores the sensitivity of CD8^+^ T cells to PD-1/PD-L1 antibodies. Preclinical studies indicate that the combination of these two agents can increase tumor volume reduction rates by over 50% in melanoma models [13]. b. Enhancing antigen presentation function. Blocking C5aR1 promotes dendritic cell (DC) maturation and enhances their antigen presentation capacity. When combined with CTLA-4 inhibitors, it can activate a broader antitumor immune response. These strategies not only open up new avenues for the treatment of complement system-related diseases, but also provide insights into addressing the challenges of specificity, efficacy, and clinical translation of targeted interventions.

As our understanding of the molecular mechanisms of the complement system in disease continues to deepen, novel targeted intervention strategies based on C5aR1 structural biology have demonstrated significant potential in the field of translational medicine. These strategies include: 1. Allosteric modulator pretreatment. Developing an oral C5aR1 allosteric modulator for preoperative administration to regulate systemic C5aR1 activity, thereby reducing the intensity of inflammatory responses during ischemia–reperfusion. This can be achieved through utilizing artificial intelligence to predict drug distribution and metabolism, optimizing the structure to enhance bioavailability and targeting. 2. Intravenous injection of nanobodies. Screening for freeze-dried powder injections of nanobodies that rapidly neutralize C5a, administered intravenously immediately before or after surgery in scenarios such as organ transplantation and myocardial infarction reperfusion. This can effectively block C5a binding to receptors, reduce inflammatory cell infiltration and tissue damage, and leverage their rapid onset of action to provide protective effects within a critical time window. 3. Dual-function molecule synergistic protection. Designing dual-function molecules that simultaneously target C5a and intracellular oxidative stress molecules, such as NADPH oxidase subunits. These molecules can be administered intravenously to inhibit complement activation and alleviate oxidative stress. It is noteworthy that the early anti-C5 mabs such as Eculizumab and Ravulizumab are also currently approved and in use, despite the fact that they are accompanied by increased susceptibility risk to microbial infections. In this context, avacopan does not block the formation of C5b and the membrane attack complex (MAC) as the C5 mabs. Furthermore, in some cases there are breakthrough cases of incomplete inhibition by eculizumab leaving residual C5 activity during strong complement activation [85]. Therefore, the disruption of the C5a/C5aR1 signalling by potent C5aR1 inhibitors, may strategically be more effective in managing complement activation in diverse disease settings.

## 9. Conclusions and Prospect

The C5a-C5aR1 pathway plays a crucial double-edged role in the progression of various diseases, but its complexity also presents a challenge for targeted interventions. With the increasing incidence of metabolic syndromes, more attention should be paid to the role of the C5a-C5aR1 pathway in inflammatory reactions and chronic diseases (such as autoimmune diseases, malignant tumors, and rheumatoid arthritis), especially the dose effect of double-edged C5a-C5aR1 pathway regulation. Moving forward, it is crucial to deepen our understanding of the complex regulatory networks of the complementary systems in different diseases, foster interdisciplinary collaboration, and incorporate advanced technologies such as single-cell sequencing, gene editing, structural biology, and artificial intelligence. These efforts will accelerate the development and clinical translation of novel targeted therapies.

## Figures and Tables

**Figure 1 ijms-26-11693-f001:**
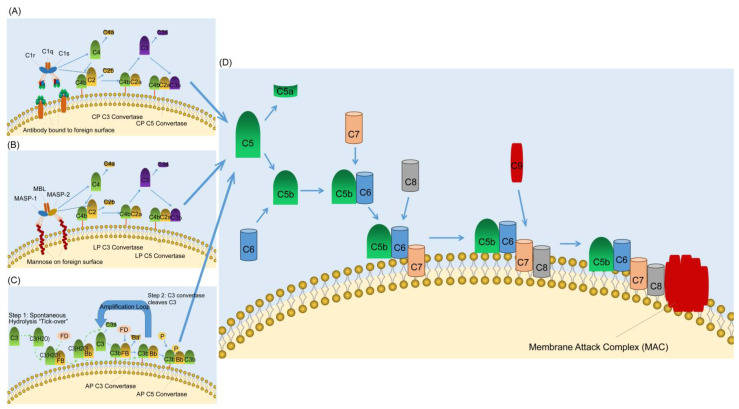
Complement activating and terminal pathways. (**A**) The classical pathway (CP) of complement activation. The CP is initiated by the binding of the C1 complex (C1q, C1r, and C1s) to bound antibody. C1r activates C1s which first cleaves C4 and then cleaves C2 leading to the formation of the CP C3 convertase (C4bC2a). The C3 convertase subsequently cleaves C3 leading to the formation of the C5 convertase (C4bC2aC3b) and the release of the anaphylatoxin C3a. The C5 convertase then cleaves C5 into C5a and C5b. C5a is the most powerful of the anaphylatoxins and C5b is the first component of the terminal pathway of complement. (**B**) The lectin pathway (LP) of complement activation. The LP is initiated by binding of the complex of mannose-binding lectin (MBL) and the serine proteases mannose-binding lectin associated proteases 1 and 2 (MASP-1 and MASP-2) to mannose groups on the surface of invading pathogens. Next, MASP-1 activates MASP-2 which acts like C1s in the classical pathway and lead to the formation of the C3 convertase. The remaining steps are the same as in the CP. (**C**) The alternative pathway of complement activation. The alternative pathway is initiated by the spontaneous hydrolysis of C3 and to form C3(H_2_O). Then C3(H_2_O) in the fluid phase binds factor B (FB), which is subsequently cleaved by factor D (FD). The fluid phase C3 convertase C3(H_2_O)Bb is formed, and it is this that cleaves C3 to C3a and C3b (Green dashed arrow). The C3 convertase cleaves C3 releasing more C3a and resulting in the formation of the alternative pathway C5 convertase (C3bBbC3b). (**D**) The terminal pathway of complement. All three pathways of complement activation converge in the terminal pathway. The terminal pathway begins when C5 is cleaved into C5a and C5b by C5 convertase. C5a is the most powerful of the anaphylatoxins and C5b binds to C6. C7 then binds to the C5bC6 complex and the newly formed C5bC6C7 complex inserts into the target membrane. C8 subsequently binds to the C5bC6C7 complex and creates a small pore in the target membrane. The final step in the terminal pathway is the binding of C9 molecules (up to 18) to the C5bC6C7C8 complex forming the membrane attack complex. The blue arrows indicate the regulation processes.

**Figure 2 ijms-26-11693-f002:**
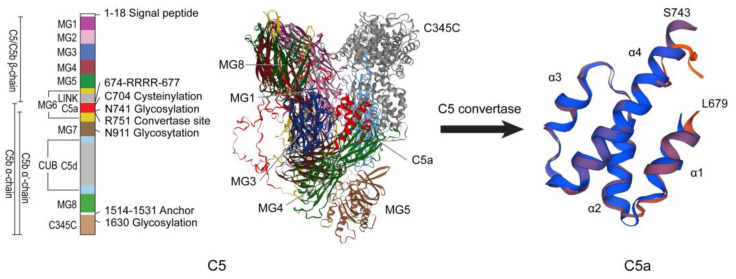
Structures of C5 and C5a. The structures of C5 and C5a were repainted according to Laursen et al. [27]. The three-dimensional structures of C5 and C5a were modeled through SWISS-MODEL (https://swissmodel.expasy.org/, accessed on 17 September 2024) based on the human C5 protein sequence (NP_001304092.1).

## Data Availability

No new data were created or analyzed in this study. Data sharing is not applicable to this article.

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
