# Peer review of "The Pleiotropic Effect of Complement C5a-C5aR1 Pathway in Diseases: From Immune Regulation to Targeted Therapy"

_ijms, 2025, doi:10.3390/ijms262311693_

Round 1
Reviewer 1 Report
Comments and Suggestions for Authors
You have written an interesting review on a timely topic. However, the manuscript contains multiple factual and terminological inaccuracies. In addition, I believe that the manuscript could be significantly improved by expanding some sections and slightly refining the structure.
- First of all, I would suggest making the background more solid.
1.1. Figure S1 is better moved to the main text, and the Supplementary material is omitted. However, there are comments to the figure itself and to the caption (see below).
1.2. I suggest that a brief background of anaphylatoxins in general and their receptors be given in the Introduction.
1.3. The fragment describing the regulation of anaphylatoxins by carboxypeptidases is better moved from section 8 to the Introduction as well.
- You consider in detail the contribution of the C5a-C5aR1 axis to a number of pathologic processes. Your choice and interest in certain pathologies is not objectionable, but it would be correct to expand the list by simply stating in one sentence other pathologies in which C5a and C5aR1 are involved.
- I suggest expanding section 8. You only describe the general principles of possible therapeutic approaches and mention only two C5aR1 antagonists (PMX-53 and avdoralimab). I think more examples should be given, perhaps with brief structural characterizations as well as information on clinical trials where relevant.
Some formal remarks apply to the manuscript as a whole.
- Some abbreviations in the text and figure captions are not deciphered. This is justified when commonly known abbreviations are used, but a balance must be maintained. You have not deciphered FSGS, MDSCs, MEK1, RAS, RAF1 and B-RAF, TAM. On the other hand, you deciphered CXCL, PKA, PLC, the meaning of which, as it seems to me, is more widely known than most of the abbreviations mentioned above. By the way, you deciphered PKA wrongly (correctly - protein kinase A).
- Along with C5aR1 and C5aR2, you often use the designation C5aR. It most likely means C5aR1 as it has been referred to previously. If so, it is best to maintain the uniform designation. If you feel it is appropriate to keep the designation C5aR, you should specify in the Introduction in what meaning you use it.
- Please carefully check that the citation is correct. I have not checked this intentionally, but I have occasionally consulted sources from the References when I wanted to clarify some information. At a minimum, your references 19, 20, and 34 are irrelevant. Number 19 should be this article: 10.1016/j.celrep.2022.110851.
Comments on certain portions of the manuscript.
- Line 74. C5a may be referred to as a large peptide or a small protein, but not a small peptide. To be quite precise, C5a is a glycosylated peptide.
- Lines 105-107. Macrophages are repeated twice.
- Lines 108-109. To talk about synergistic activation, you need at least two activating agents. In your example, C5a and LPS together act synergistically (or we could say that C5a acts synergistically with LPS). However, C5a does not increase LPS production, as your text suggests.
- Line 116. Sepsis is not a disease.
- Figure 2 and caption (lines 124-125). In the text of the caption cAMP activates PLC, in the figure it inhibits. In fact, there is no direct interaction between them.
- Lines 246-247. It is better to write ‘antagonists’ in full, not to use an abbreviation.
- Lines 340-342. Again, if LPS activates complement synergistically, at least one other activating agent must be mentioned. In addition, LPS activates the alternative pathway of complement without the involvement of TLR4.
- Lines 375-376. PMX-53 is a peptide and avdoralimab is a monoclonal antibody. Both are not related to small molecules.
- Lines 377-379. Perhaps you meant C5aR, not C3aR?
- Line 397. Usually written C5a-desArg or less frequently C5a desArg.
- Figure S1. C1s is incorrectly labeled as C1r, and C6 is incorrectly labeled as C8.
- Figure S1 and caption. Activation of the alternative pathway is incorrectly described and depicted in the figure. Tick-over implies spontaneous hydrolysis of the thioester rather than the peptide bond in the C3 molecule. The cleavage of C3 into C3a and C3b in the course of spontaneous activation of the alternative pathway occurs subsequently by the action of liquid-phase C3 convertase.
- Ibid. There is no unambiguous convention on what to consider as the start of the terminal pathway, the cleavage of C5 or the interaction of C5b with C6. You can choose either option as you wish. However, in the figure you have one option and in the caption text you have another.
- Ibid. A natural membrane attack complex includes no more than 18 C9 molecules (not 21). A poly-C9 transmembrane pore (i.e., a pore formed in vitro from C9 without C5b-8) can include up to 22 C9 molecules.
Author Response
Responses to Comments and Suggestions for Authors
Comment
- First of all, I would suggest making the background more solid.
1.1. Figure S1 is better moved to the main text, and the Supplementary material is omitted. However, there are comments to the figure itself and to the caption (see below).
Response
We have moved the Figure S1 to the main text and revised the figure and caption according to your comment.
Comment
1.2. I suggest that a brief background of anaphylatoxins in general and their receptors be given in the Introduction.
Response
We have given a brief background of anaphylatoxins and their receptors in general according to your comment.
Comment
1.3. The fragment describing the regulation of anaphylatoxins by carboxypeptidases is better moved from section 8 to the Introduction as well.
Response
Thank you very much for your comment. We have moved the fragment describing the regulation of anaphylatoxins by carboxypeptidases to Introduction section according to your comment.
Comment
- You consider in detail the contribution of the C5a-C5aR1 axis to a number of pathologic processes. Your choice and interest in certain pathologies is not objectionable, but it would be correct to expand the list by simply stating in one sentence other pathologies in which C5a and C5aR1 are involved.
Response
We completely agree with your comment. However, it seems that the C5a-C5aR1 pathway is involved in almost all immune, infectious, and metabolic diseases, and is also associated with some degenerative diseases, making it difficult to summarize in a single sentence exactly which diseases it plays a role in.
Comment
- I suggest expanding section 8. You only describe the general principles of possible therapeutic approaches and mention only two C5aR1 antagonists (PMX-53 and avdoralimab). I think more examples should be given, perhaps with brief structural characterizations as well as information on clinical trials where relevant.
Response
Thank you for your comment. We added some other study according to your comment.
Comment
Some formal remarks apply to the manuscript as a whole.
- Some abbreviations in the text and figure captions are not deciphered. This is justified when commonly known abbreviations are used, but a balance must be maintained. You have not deciphered FSGS, MDSCs, MEK1, RAS, RAF1 and B-RAF, TAM. On the other hand, you deciphered CXCL, PKA, PLC, the meaning of which, as it seems to me, is more widely known than most of the abbreviations mentioned above. By the way, you deciphered PKA wrongly (correctly - protein kinase A).
Response
Thank you for your comment. We have deciphered the abbreviations. Moreover, we have revised the description of PKA according to your comment.
Comment
- Along with C5aR1 and C5aR2, you often use the designation C5aR. It most likely means C5aR1 as it has been referred to previously. If so, it is best to maintain the uniform designation. If you feel it is appropriate to keep the designation C5aR, you should specify in the Introduction in what meaning you use it.
Response
Thank you very much for your comment. Unless otherwise specified, it should be C5aR1, and we have modified it according to your comments.
Comment
- Please carefully check that the citation is correct. I have not checked this intentionally, but I have occasionally consulted sources from the References when I wanted to clarify some information. At a minimum, your references 19, 20, and 34 areirrelevant. Number 19 should be this article: 10.1016/j.celrep.2022.110851 .
Response
Thank you for your comment. We have carefully rechecked and revised the references according to your comment.
Comment
Comments on certain portions of the manuscript.
- Line 74. C5a may be referred to as a large peptide or a small protein, but not a small peptide. To be quite precise, C5a is a glycosylated peptide.
Response
Thank you very much for your comment. We have revised expression of C5a according to your comment.
Comment
- Lines 105-107. Macrophages are repeated twice.
Response
Thank you for your comment. We have deleted the repeated sentence.
Comment
- Lines 108-109. To talk about synergistic activation, you need at least two activating agents. In your example, C5a and LPS together act synergistically (or we could say that C5a acts synergistically with LPS). However, C5a does not increase LPS production, as your text suggests.
Response
Thank you for your comment. We have revised the expression according to your comment.
Comment
- Line 116. Sepsis is not a disease.
Response
Thank you for your comment. We have deleted sepsis to avoid this error.
Comment
- Figure 2 and caption (lines 124-125). In the text of the caption cAMP activates PLC, in the figure it inhibits. In fact, there is no direct interaction between them.
Response
We have revised the caption and figure according to your comment.
Comment
- Lines 246-247. It is better to write ‘antagonists’ in full, not to use an abbreviation.
Response
Thank you for your comment. We have revised to the antagonists in full according to your comment.
Comment
- Lines 340-342. Again, if LPS activates complement synergistically, at least one other activating agent must be mentioned. In addition, LPS activates the alternative pathway of complement without the involvement of TLR4.
Response
Thank you for your comment. We have removed the incorrect statements based on your comments.
Response
- Lines 375-376. PMX-53 is a peptide and avdoralimab is a monoclonal antibody. Both are not related to small molecules.
Response
We have revised the expression according to your comment.
Comment
- Lines 377-379. Perhaps you meant C5aR, not C3aR?
Response
Thank you for your comment. It should be C5aR. We have revised the errors.
Comment
- Line 397. Usually written C5a-desArg or less frequently C5a desArg.
Response
Thank you for your comment. We have revised the C5a desArg to C5a-desArg.
Comment
- Figure S1. C1s is incorrectly labeled as C1r, and C6 is incorrectly labeled as C8.
Response
Thank you very much for pointing out the errors. We have revised the errors according to your comment.
Comment
- Figure S1 and caption. Activation of the alternative pathway is incorrectly described and depicted in the figure. Tick-over implies spontaneous hydrolysis of the thioester rather than the peptide bond in the C3 molecule. The cleavage of C3 into C3a and C3b in the course of spontaneous activation of the alternative pathway occurs subsequently by the action of liquid-phase C3 convertase.
Response
Thank you very much for your comment. Since the activation of the alternative pathway is divided into to processes: spontaneous hydrolysis and subsequent C3 convertase action, we modified the image to better illustrate this process.
Comment
- Ibid. There is no unambiguous convention on what to consider as the start of the terminal pathway, the cleavage of C5 or the interaction of C5b with C6. You can choose either option as you wish. However, in the figure you have one option and in the caption text you have another.
Response
Thank you for your comment. We have corrected the inconsistencies in the figure caption text according to your comment.
Comment
- Ibid. A natural membrane attack complex includes no more than 18 C9 molecules (not 21). A poly-C9 transmembrane pore (i.e., a pore formed in vitro from C9 without C5b-8) can include up to 22 C9 molecules.
Response
Thank you for your comment. We have revised the error.
Reviewer 2 Report
Comments and Suggestions for Authors
I am writing to thank you very much for providing me with the opportunity to evaluate this interesting review article. It attempts to bring together knowledge acquired over several years of work, about the roles of the Complement C5a-C5aR1 interactions in the regulation of cellular homeostasis and inflammation. This field has expanded considerably over the years as the Complement system has emerged as a principle pillar of innate immunity with multidimensional contributions to various cell interactions in immune defense, cancer, autoimmune diseases and metabolic disorders. C5a is the most potent Complement anaphylatoxin and its interaction with the C5a receptors influences many diverse processes during increase and/or sustained Complement activation. Therefore, its pharmacological targeting attracts considerable interest in various disease and pathological conditions.
Overall, this review is well written in terms of flow and discussion and cites many importnant articles. It is worth getting published, however prior to that i have a number of kind requests by the authors:
- I recently came across a great study investigating the contribution of C5a in renal injury using an LPS-based mouse model (Complement C5aR blockade attenuates LPS-induced acute kidney injury by regulating ferroptosis via nuclear factor-erythroid 2-related factor 2 signaling in mice, https://doi.org/10.1016/j.freeradbiomed.2025.07.021). This study investigates mechanistically the contribution of increased C5a-C5aR induced-signalling in the induction of ferroptosis in tubular epithelial cells via the p38/Nrf2/SLC7A11/GPX4 signaling pathway. In this study, C5aR blockade with the C5aR antagonist W54011 (small molecule) reversed LPS-induced acute kidney injury and attenuated mitochondrial damage in tubular epithelial cells. This therapeutic blockage was accompanied with reduced NAG levels and proinflammatory cytokines, as well as increased oxidative stress-related markers. I think that you should try discussing these findings in sections 3.2 (around lines 218-219) and 4 (around lines 242-243) and possibly the C5aR antagonist W54011 in section 8 of your study.
- In section 2 of your study it would worth adding some extra information on C5aR2. There is some very detailed information in section 7 of the review article that you cite in ref. 21.
- Please increase the very small fonts in the left panel of Fig. 1 and upper panel of Fig. 2. In the latter please correct to moderate levels.
- Across your text please add explain the abbreviations: MDSCs (Myeloid-derived suppressor cells), Lines 22, 196, 197 and 376; TAMs (Tumor-associated macrophages), Lines 200.
- After reading your affiliation with the Hunan Agricultural University and the Featured Aquatic Resources Utilization, i would like to ask you whether you would like to discuss in section 8 the potential pharmacological C5a-C5aR blockage intervention as part of certain thereapeutic strategies for livestock or fisheries applications. Could that be achieved with a small molecule inhibitor of C5aR and be useful in certain inflammatory conditions along with other therapeutics (e.g. antimicrobials, antibiotics, antivirals, anticoagulants etc)?
Author Response
Responses to Comments and Suggestions for Authors Comment 1. I recently came across a great study investigating the contribution of C5a in renal injury using an LPS-based mouse model (Complement C5aR blockade attenuates LPS-induced acute kidney injury by regulating ferroptosis via nuclear factor-erythroid 2-related factor 2 signaling in mice, https://doi.org/10.1016/j.freeradbiomed.2025.07.021). This study investigates mechanistically the contribution of increased C5a-C5aR induced-signalling in the induction of ferroptosis in tubular epithelial cells via the p38/Nrf2/SLC7A11/GPX4 signaling pathway. In this study, C5aR blockade with the C5aR antagonist W54011 (small molecule) reversed LPS-induced acute kidney injury and attenuated mitochondrial damage in tubular epithelial cells. This therapeutic blockage was accompanied with reduced NAG levels and proinflammatory cytokines, as well as increased oxidative stress-related markers. I think that you should try discussing these findings in sections 3.2 (around lines 218-219) and 4 (around lines 242-243) and possibly the C5aR antagonist W54011 in section 8 of your study. Response Thank you very much for your recommendation. The study provides an excellent example for the treatment of LPS-induced diseases through blocking C5R pathway. Although this example also has certain reference value in the prevention and treatment of tumor and fibrosis, we think it is more suitable to discuss it in the therapy research, so we cite this study in the section 8. Comment 2. In section 2 of your study it would worth adding some extra information on C5aR2. There is some very detailed information in section 7 of the review article that you cite in ref. 21. Response Thank you for your comment. We have added some information on C5aR2 in the section 2 according to your comment. Comment 3. Please increase the very small fonts in the left panel of Fig. 1 and upper panel of Fig. 2. In the latter please correct to moderate levels. Response We have revised the figures according to your comment. Comment 4. Across your text please add explain the abbreviations: MDSCs (Myeloid-derived suppressor cells), Lines 22, 196, 197 and 376; TAMs (Tumor-associated macrophages), Lines 200. Response Thank you for your comment. We have added the explains of the abbreviations when they first appeared. Comment 5. After reading your affiliation with the Hunan Agricultural University and the Featured Aquatic Resources Utilization, i would like to ask you whether you would like to discuss in section 8 the potential pharmacological C5a-C5aR blockage intervention as part of certain thereapeutic strategies for livestock or fisheries applications. Could that be achieved with a small molecule inhibitor of C5aR and be useful in certain inflammatory conditions along with other therapeutics (e.g. antimicrobials, antibiotics, antivirals, anticoagulants etc)? Response Thank you very much for your attention to the progress of our research. Although we have previously studied the role of the C5a-C5aR1 pathway in pathogen infections in fish, these studies are still at an early stage. One of the purposes of this review is also to explore potential strategies for treating infectious diseases in fish thorough summarizing current research on C5a-C5aR1 and combing it with our findings. However, given that the research is still preliminary, we do not yet have ideal recommendations or ideas for applying the regulation of the C5a-C5aR pathway to prevent infectious diseases in fish. Nevertheless, we believe it is a promising approach worth exploring, and we will further investigate it in future studies. Therefore, in this review, we have not included potential pharmacological C5a-C5aR blockage interventions as part of certain therapeutic strategies for livestock or fisheries applications.Round 2
Reviewer 1 Report
Comments and Suggestions for Authors
You have addressed most of the comments and significantly improved the manuscript. However, an inaccuracy remains in Figure 1. Spontaneous activation of the alternative pathway is a bit more complex than you represent. First, spontaneous hydrolysis is the hydrolysis of the thioester bond in the C3 molecule to form C3(Hâ‚‚O) (and not C3a and C3b). Then C3(H2O) in the fluid phase binds FB, which is subsequently cleaved by FD. Thus, the fluid phase C3 convertase C3(H2O)Bb is formed, and it is this that cleaves C3 to C3a and C3b.
Author Response
Comment
You have addressed most of the comments and significantly improved the manuscript. However, an inaccuracy remains in Figure 1. Spontaneous activation of the alternative pathway is a bit more complex than you represent. First, spontaneous hydrolysis is the hydrolysis of the thioester bond in the C3 molecule to form C3(Hâ‚‚O) (and not C3a and C3b). Then C3(H2O) in the fluid phase binds FB, which is subsequently cleaved by FD. Thus, the fluid phase C3 convertase C3(H2O)Bb is formed, and it is this that cleaves C3 to C3a and C3b.
Response
Thank you very much for your comment. We have revised the spontaneous activation of the alternative pathway in Figure 1 according to your comment.
Reviewer 2 Report
Comments and Suggestions for Authors
Thank you very much for providing me with the opportunity to re-evaluate this interesting review article on the biology and the pleiotropic effects of the C5a/C5aR1 interaction.
I read the revised version and i noticed that the authors took into consideration the suggestions and points raised during the previous round. I think that the paper now is more enriched and reads well.
However, i still have some additional comments after reading the text:
-Line 93: '...on the surface of activating surfaces (non-host surfaces) and the release of C3a'. In terms of scientific accuracy you should probably add in the parenthesis the altered host surface, i.e. damaged or compromised host cell surface that may occur in different circumstances. Although such cases may vary among pathologies, they can include events of membrane rupture due to uncontrolled oxidative cell stress during hemolytic events or non-canonical alternative pathway activation during viral surface engagement and cellular infection. So, a revised version could read as: '...on the surface of activating surfaces (altered or compromised host cell surfaces and non-host surfaces) and the release of C3a'.
-Line 400: 8. Current status of C5a/C5aR1 targeted therapy research. In section 8 i noticed that you added additional information refering to the recent W54011-based work exploring the contribution of the C5a/C5aR pathway to mitochondrial dysfunction, ferroptosis and sepsis after LPS challenge. This is a very important section of your review and after reading it, i revisited the literature to explore further the existing literature.
There is an excellent 2020 article (https://doi.org/10.1016/j.bcp.2020.114156) that examines and compares all the existing peptidic and non-peptide C5aR inhibitors including the W54011 and the approved CCX168 (avacopan). You mention the peptidic PMX53 in section 8, but i think that you should make the effort to expand and restucture this section to add available information and cite some additional information. This section is very important, because the approved CCX168 (avacopan/TAVNEOS), could potentially be utilized for additional indications and pathologies that you discuss in your review. Either alone or perhaps along additional next gen AP complement inhibitors such as danicopan (CFD inhibition) or iptacopan (CFB inhibitors) (https://doi.org/10.1111/imr.13143) and potential others targeting non-complement components as you mention (e.g. corticosteroids, checkpoint inhibitors as well as others).
Also in this section you mention aC5a nanobodies, but you have to keep in mind that the early anti-C5 mabs such as Eculizumab and Ravulizumab are also currently apporoved and in use, despite the fact that they are accompanied by increased susceptibility risk to microbial infections. In this context, Avacopan does not block the formation of C5b and the membrane attack complex (MAC) as the C5 mabs. Also in some cases there are breakthrough cases of incomplete inhibition by eculizumab leaving residual C5 activity during strong complement activation (Incomplete inhibition by eculizumab: mechanistic evidence for residual C5 activity during strong complement activation (https://doi.org/10.1182/blood-2016-08-732800).
In this context therefore, the disruption of the C5a/C5aR1 signalling by potent aC5aR1 inhibitors, may strategically be more effective in managing complement activation in diverse disease settings.
Author Response
Comment
-Line 93: '...on the surface of activating surfaces (non-host surfaces) and the release of C3a'. In terms of scientific accuracy you should probably add in the parenthesis the altered host surface, i.e. damaged or compromised host cell surface that may occur in different circumstances. Although such cases may vary among pathologies, they can include events of membrane rupture due to uncontrolled oxidative cell stress during hemolytic events or non-canonical alternative pathway activation during viral surface engagement and cellular infection. So, a revised version could read as: '...on the surface of activating surfaces (altered or compromised host cell surfaces and non-host surfaces) and the release of C3a'.
Response
Thank you very much for your comment. We have revised the sentence according to the another reviewer’s and your comments.
Comment
-Line 400: 8. Current status of C5a/C5aR1 targeted therapy research. In section 8 i noticed that you added additional information refering to the recent W54011-based work exploring the contribution of the C5a/C5aR pathway to mitochondrial dysfunction, ferroptosis and sepsis after LPS challenge. This is a very important section of your review and after reading it, i revisited the literature to explore further the existing literature.
There is an excellent 2020 article (https://doi.org/10.1016/j.bcp.2020.114156 ) that examines and compares all the existing peptidic and non-peptide C5aR inhibitors including the W54011 and the approved CCX168 (avacopan). You mention the peptidic PMX53 in section 8, but i think that you should make the effort to expand and restucture this section to add available information and cite some additional information. This section is very important, because the approved CCX168 (avacopan/TAVNEOS), could potentially be utilized for additional indications and pathologies that you discuss in your review. Either alone or perhaps along additional next gen AP complement inhibitors such as danicopan (CFD inhibition) or iptacopan (CFB inhibitors) (https://doi.org/10.1111/imr.13143 ) and potential others targeting non-complement components as you mention (e.g. corticosteroids, checkpoint inhibitors as well as others).
Response
Thank you very much for recommending such an excellent research paper. We have supplemented the content of this study in section 8 and added relevant research on CCX168. Additionally, we have reorganized the structure of the section 8 according to your comments.
Comment
Also in this section you mention aC5a nanobodies, but you have to keep in mind that the early anti-C5 mabs such as Eculizumab and Ravulizumab are also currently apporoved and in use, despite the fact that they are accompanied by increased susceptibility risk to microbial infections. In this context, Avacopan does not block the formation of C5b and the membrane attack complex (MAC) as the C5 mabs. Also in some cases there are breakthrough cases of incomplete inhibition by eculizumab leaving residual C5 activity during strong complement activation (Incomplete inhibition by eculizumab: mechanistic evidence for residual C5 activity during strong complement activation (https://doi.org/10.1182/blood-2016-08-732800) .
In this context therefore, the disruption of the C5a/C5aR1 signalling by potent aC5aR1 inhibitors, may strategically be more effective in managing complement activation in diverse disease settings.
Response
Thank you very much for your insightful comments. We have supplemented the contents in the section 8, and we sincerely appreciate your truly perceptive comments.